# Clinical Effectiveness of Restorative Materials for the Restoration of Carious Primary Teeth: An Umbrella Review

**DOI:** 10.3390/jcm11123490

**Published:** 2022-06-17

**Authors:** Stefanie Amend, Kyriaki Seremidi, Dimitrios Kloukos, Katrin Bekes, Roland Frankenberger, Sotiria Gizani, Norbert Krämer

**Affiliations:** 1Department of Paediatric Dentistry, University Medical Centre Giessen and Marburg (Campus Giessen), Medical Centre for Dentistry, Justus-Liebig-University Giessen, Schlangenzahl 14, 35392 Giessen, Germany; norbert.kraemer@dentist.med.uni-giessen.de; 2Department of Paediatric Dentistry, Athens School of Dentistry, National & Kapodistrian University of Athens, 2 Thivon Str, Goudi, 115 27 Athens, Greece; kser51@otenet.gr (K.S.); sotiriagizani@gmail.com (S.G.); 3Department of Orthodontics and Dentofacial Orthopedics, School of Dental Medicine, University of Bern, Freiburgstrasse 7, 3010 Bern, Switzerland; dimitrios.kloukos@zmk.unibe.ch; 4Department of Paediatric Dentistry, University Clinic of Dentistry, Medical University Vienna, Sensengasse 2a, 1090 Vienna, Austria; katrin.bekes@meduniwien.ac.at; 5Department of Operative Dentistry, Endodontics and Paediatric Dentistry, University Medical Centre Giessen and Marburg (Campus Marburg), Medical Centre for Dentistry, Philipps-University Marburg, Georg–Voigt–Str. 3, 35039 Marburg, Germany; frankbg@med.uni-marburg.de

**Keywords:** restorative materials, biomaterials, primary teeth, dental caries, umbrella review

## Abstract

Since untreated dental caries remain a worldwide burden, this umbrella review aimed to assess the quality of evidence on the clinical effectiveness of different restorative materials for the treatment of carious primary teeth. A literature search in electronic bibliographic databases was performed to find systematic reviews with at least two-arm comparisons between restorative materials and a follow-up period ≥12 months. Reviews retrieved were screened; those eligible were selected, and the degree of overlap was calculated using the ‘corrected covered area’ (CCA). Data were extracted and the risk of bias was assessed using the ROBIS tool. Fourteen systematic reviews with a moderate overlap (6% CCA) were included. All materials studied performed similarly and were equally efficient for the restoration of carious primary teeth. Amalgam and resin composite had the lowest mean failure rate at 24 months while high-viscosity and metal-reinforced glass ionomer cements had the highest. At 36 months, high-viscosity glass ionomer cements showed the highest failure rate with compomer showing the lowest. Most reviews had an unclear risk of bias. Within the limitations of the review, all materials have acceptable mean failure rates and could be recommended for the restoration of carious primary teeth.

## 1. Introduction

Despite the notable decline in caries over recent years [1], millions of untreated carious primary teeth still constitute a global burden and a major health care challenge [2,3]. There are several treatment options for carious primary teeth with increasing invasiveness ranging from minimum intervention dentistry to restorative approaches [4,5]. Conservative caries management strategies aim to sustain hard dental tissue and to prolong the cycle of dental re-restoration, both of which increase the tooth’s long-term preservation in the oral cavity [6]. However, it is recommended to restore carious lesions presenting with cavitated surfaces that cannot be cleaned properly or sealed in order to reduce the lesion’s caries activity [6].

When it comes to the restoration of primary teeth, several factors may influence longevity, such as the operator’s experience, child’s age, type of tooth, cavity size, or isolation technique [7]. Age-related restrictions in the cooperation of the child [8], the individual caries risk [9], and the limited lifespan of primary teeth due to physiological exfoliation [10] need to be taken into consideration during treatment planning. Apart from the operator- and patient-related factors, the choice of restorative material may influence the achieved outcome with varying annual failure rates of restorations in primary teeth, depending on the material under investigation [8].

Amalgam has been used in Paediatric Dentistry for decades to restore carious primary molars [10,11,12,13,14], although its preparation is more invasive since a retentive cavity design is needed, causing a higher loss of hard dental tissue [10,13]. With the implementation of the Minamata Convention on Mercury in 2013, it has been attempted “to phase down” the use of mercury and mercury-containing products, including dental amalgam, to avoid future environmental pollution [15].

Alternatively, different glass-ionomer cements (GICs) [10,12], compomers [11,13], or resin composites [7,14] are placed directly in the cavity; or primary teeth are restored indirectly with stainless steel or tooth-coloured crowns [16]. In general, glass-ionomer cements bind chemically to the dental hard tissue, release fluoride, can be placed in bulk (reducing treatment time), and are less technique-sensitive than adhesively bonded restorative materials. However, both the type of GIC chosen and the cavity class are relevant [17]. Conventional GICs and metal-reinforced glass-ionomer cements (MRGIC) show inferior mechanical properties, resulting in increased wear and fractures when used in load-bearing posterior teeth, especially in Class–II cavities, reducing survival time [10,18]. To improve fracture resilience and wear resistance, GICs were modified by adding methacrylate components [17], enabling a more favourable clinical success rate of such resin-modified glass-ionomer cements (RMGIC) [19]. Furthermore, high-viscosity GICs (HVGICs) were introduced and have been used successfully for the Atraumatic Restorative Treatment (ART) of primary molars [20].

Adhesive restorations with compomer or resin composite are more technique-sensitive and require a longer treatment time [17]. For primary molars with multi-surface lesions, stainless steel crowns and the more recently introduced zirconia crowns offer an opportunity to cover the whole tooth crown, with a similar clinical performance of over three years [16,17].

Considering that many clinical studies evaluated the survival of the restorative treatment approaches mentioned above, the question arises which restorative material is the most suitable for carious primary teeth. This umbrella review aims to assess the quality of evidence of published systematic reviews on the clinical effectiveness of different restorative materials, including new biomaterials, for the restoration of carious primary teeth in children at least 12 months of age.

## 2. Materials and Methods

### 2.1. Registration and Reporting Format

This review was registered in the PROSPERO international prospective register of systematic reviews hosted by the National Institute for Health Research (NIHR), University of York, UK, Centre for Reviews and Dissemination (CRD42021227889). 

The umbrella review was conducted in accordance with the Cochrane methodology and the Preferred Reporting Items for Systematic Reviews and Meta-analysis (PRISMA) was adopted throughout the process [21].

### 2.2. Inclusion Criteria

Systematic reviews with or without meta-analyses, reporting on restorative materials for the restoration of carious primary teeth, were included. 

Reviews were included which were initial studies of any design:Comparing different dental materials for the restoration of carious lesions in primary teeth (the use of local anaesthesia and/or rubber dam isolation was not a limitation);With a clear definition of lesion location, size, and depth;With asymptomatic teeth (no history of pain, pulp exposure, infection, swelling, or evidence of periapical pathology);With at least two-arm comparisons;With a follow-up period of at least 12 months.

### 2.3. Exclusion Criteria

Reviews reporting: treatment of caries in permanent teeth (including those where results on primary dentition were not reported separately), with a single cohort of patients, management of superficial caries (i.e., radiographic lesions R1), and with a follow-up period <12 months were excluded.

### 2.4. PICO Format for the Included Studies

**P**articipants: Children up to the age of 12 years with deep carious lesions in their primary dentition, with no restrictions on participants’ demographic characteristics.**I**ntervention: Any dental material placed as restoration in primary teeth: amalgam (A), resin composite (RC), compomer (CO), GIC, RMGIC, metal-reinforced GIC (MRGIC), HVGIC, and stainless steel crowns (SSC).**C**omparator group: Any of the above restorative materials.**O**utcomes: Primary outcomes were treatment failure/success (criteria used were not a limitation) and restoration quality (surface roughness, colour match, marginal integrity, tooth integrity, filling integrity, proximal contact, change of sensitivity, hypersensitivity, and radiographic assessment) [22].

Secondary outcomes were time until restoration failure occurs or re-treatment is needed, discomfort during restorative treatment or within 24 h after treatment, patient’s and/or carer’s perceptions of the restorative treatment, and the impact of the tooth or technique-related factors {e.g., surface(s) affected (single- or multi-surface lesions), the technique of carious removal (selective vs. complete), type of tooth (anterior/posterior tooth), isolation technique, and the type of adhesive (self-etch, etch-and-rinse, and universal)} on the clinical effectiveness of the restorative materials.

### 2.5. Search Strategy

A literature search was conducted (7 October 2021) in electronic databases (Embase, Web of Science, Dentistry & Oral Sciences Source (DOSS), Medline/PubMed, Scopus, LILACS, and The Cochrane Library (Cochrane Database of Systematic Reviews, Cochrane Methodology Register) considering the differences in controlled vocabulary and syntax rules. Reference lists of retrieved articles were hand-searched to identify additional studies not identified through electronic searches. No language or publication year restrictions were applied.

### 2.6. Data Collection and Analysis

The titles and/or abstracts of all studies retrieved from the search, and those from additional sources, were screened independently by two review authors (KS and SA) to identify those that potentially met the inclusion criteria. Full texts of the potentially included studies were reviewed and the list of included studies was finalised. Any discrepancies and disagreements were resolved through discussion with a third author (SG). All reviewers were calibrated prior to the initiation of the study, with intra- and inter-examiner reliability values being excellent, exceeding 0.85 for all cases.

Data were extracted independently and in duplicate by two review authors (KS and SA). For each review, the following data were recorded: a. publication details (authors and year of publication), b. review methodology (search strategy, objectives, number of included studies, and study design), c. review characteristics (interventions, control groups, and quantitative synthesis), and d. outcomes (main results and conclusions) including methods of assessment and quality assessment (risk of bias assessment tool used and method of grading the quality). 

For studies with missing/unclear reporting, an attempt was made to contact the authors for further clarifications. 

Numerical results reported were presented as percentage values (mean, minimum, and maximum), calculated from the relevant proportions of cases that did not fail as reported in each review. Odds ratios (OR) or risk ratios (RR) reported in the meta-analyses of each review were also recorded, where applicable. 

### 2.7. Analysis of the Degree of Overlap

The degree of overlap of primary studies in the included systematic reviews was evaluated by calculating the ‘corrected covered area’ (CCA) and generating the citation matrix [23]. CCA was calculated using the formula: CCA = N – r/rc – r, where N = number of included publications (including double counting), r = number of rows (number of index publications), and c = number of columns (number of reviews) in the citation matrix. The degree of overlap is interpreted as: 0% to 5% (slight overlap), 6% to 10% (moderate overlap), 11% to 15% (high overlap), and above 15% (very high overlap).

### 2.8. Estimation of a Common Effect Size

The estimation of size effects across all factors under investigation and the application of common effect size for all factors depending on the design and analytical approach of the studies were planned through a conversion of all effect sizes into equivalent Odds Ratios, using the RevMan 5 Software (Review Manager–RevMan {Computer program}, Version 5.4, The Cochrane Collaboration, London, UK, 2020).

### 2.9. Risk of Bias Assessment

Quality assessment was performed, independently by two review authors, using Risk of Bias in Systematic Reviews (ROBIS) [24]. The ROBIS tool critically appraises the review through specific questions in four distinct domains: a. eligibility criteria, b. identification and selection of studies, c. data collection and study appraisal, and d. synthesis and findings. In the end, the overall confidence of the review is rated (from high to critically low) through the identification of critical and non-critical weaknesses [25].

## 3. Results

### 3.1. Literature Search Results

Initial searches yielded 175 systematic reviews (Appendix A). After duplicate removal and the addition of 54 studies from hand searching, 225 reviews were considered. Following title and abstract screening, 117 were excluded. Of these 108 reviews considered eligible for inclusion, 94 were excluded after full-text assessment (Appendix A), leaving a total sample of 14 reviews that were finally included (Figure 1, according to the PRISMA statement 2020 [21]).

### 3.2. Overlap of Studies

Within the 14 reviews, 179 studies altogether were included without considering overlap (Appendix A). CCA was calculated at 0.06 (6%), with ‘N’ being equal to 179, ‘r’ equal to 101, and ‘c’ equal to 14. Overlap was moderate, with a moderate number of studies appearing several times within several reviews, slightly increasing the weighing of the results.

### 3.3. Study Characteristics

Literature searches in most reviews were performed within MEDLINE, Science–Direct, Web of Science, Google Scholar, Cochrane, and Embase databases, with two reviews [20,26] restricted to MEDLINE and EMBASE and another two [27,28] solely to MEDLINE. Two reviews also explored grey literature and dissertations [29,30] and four checked the National Institutes of Health and the World Health Organization International Clinical Trials Registry for ongoing trials [29,30,31,32]. A timeframe was defined from 1966 to 2021, with one review narrowing it between 1996 and 2017 [8], one between 2012 and 2016 [20], and one reporting a 10-year time frame [32]. Most studies placed no language restrictions, apart from three reviews [28,32,33] that included studies only published in English. 

Nine reviews included only randomized control trials (RCTs), three included both RCTs and non-RCTs [8,27,32], one included both RCTs and an observational study [34] and one included nine RCTs, one longitudinal study and one study with no information regarding its design [28] (Table 1). From the available data, 80 primary studies were of split-mouth design, and 70 of parallel-arm design. One review included initial studies with a partial split-mouth design [35], where the unit of randomisation was the tooth, not the patient. Interventions in the majority of the reviews involved conventional materials used for restoring single- or multi-surface carious lesions. Two reviews reported on stainless steel crowns [8,29], six reviews reported on atraumatic restorative treatment (ART) using mainly HVGIC [20,27,28,33,36], and one reported on different adhesive systems [32]. 

Outcomes reported were success rate, failure rate, annual failure rate, major failure, clinical performance, and caries arrest and/or progression at a follow-up period that varied from 12 months to 84 months. Most reviews used a combination of widely used criteria including FDI, modified USPHS, and ART. Four reviews included studies that used the author’s own criteria [8,27,30,34], and two reviews did not report on the criteria used [29,37]. Meta-analysis was performed in most reviews except for four [26,27,32,36], with the number of initial studies included in each meta-analysis varying from 2 to 17.

### 3.4. Risk of Bias Assessment

Most reviews used the Cochrane collaboration tool for assessing the risk of bias, three reviews used criteria proposed by other investigators [26,35,36], and one did not include quality assessment for the included studies [28]. Overall, four reviews were at low risk of bias [8,29,32,33], three were at high risk [26,28,36], and the remaining had an unclear risk of bias (Table 2). Regarding study eligibility, most reviews were at low risk except for four [26,27,30,36], which restricted search criteria, reducing the comprehensiveness and immediately increasing the bias by possibly leaving out eligible reviews. 

In three reviews, concerns were raised regarding identification and selection, with the search strategy not being reproducible and the inclusion and exclusion criteria not being clearly defined [27,28,36]. Data collection and study appraisal were adequate in most reviews, except for four in which clinical and methodological heterogeneity of the included studies was either not performed adequately or was absent. Regarding synthesis of the findings, in most reviews, concerns were raised mainly due to the fact that meta-analysis was performed in studies of unclear or high risk of bias. 

### 3.5. Findings of the Reviews

Table 3 summarises the main conclusions of the included systematic reviews and Table 4 summarises the results of the meta-analysis performed. Overall, all materials performed similarly and were equally efficient for the restoration of primary teeth with deep caries. Amalgam and resin composite had the lowest mean failure rate at 24 months and RMGIC and MRGIC were the highest (Figure 2), respectively, at 36 months (Figure 3). HVGIC showed the highest failure rate with compomer showing the lowest.

#### 3.5.1. Amalgam (A)

Performance of amalgam was reported in nine reviews [8,20,26,27,31,33,34,35,36] with the mean calculated failure rate at 24 months being 11% (range: 7%–17%) and at 36 months, 21% (range: 4%–36%). It was reported that amalgam, for multi-surface cavities in primary molars, can be expected to survive a minimum of 3.5 years but potentially more than 7 years and remains an appropriate treatment option. Meta-analysis was performed and found that amalgam, compared to GIC and RMGIC restorations, in primary molars exhibited a lower failure rate but not in a statistically significant manner. The quality of the evidence was of unclear risk of bias due to methodological heterogeneity.

#### 3.5.2. Resin Composite (RC)

The mean failure rate for RC as calculated from the ten reviews reporting on its efficacy varied from 14% at 24 months to 20% at 36 months [8,20,26,27,30,31,32,34,36,37]. Resin composite exhibited the lowest failure rates, with all resin-based restorative materials (resin composite and compomer) having no statistically significant differences based on a “moderate” level of evidence. In the review of Delgado et al. (2021) it was reported that innovative materials, such as bulk-fill resin composites, self-adhesive restoratives, and adhesives, perform comparably to conventional restorative materials and all are clinically acceptable [32].

#### 3.5.3. Glass-ionomer Cements (GICs)

A total of 13 studies reported on GIC in various forms with the calculated mean failure rate at 24 months ranging between 16% and 21% and 16% and 35% at 36 months. At both time intervals, the lowest failure rates were reported with RMGIC and the highest with HVGIC for the restoration of primary teeth. Overall, results are inconclusive, with most reviews [20,26,27,29,30,31,33,34,35,36,37] reporting a similar or even better performance of GICs when compared to conventional restorative materials. Characteristically, Dias et al. (2018) and Mickenautsch et al. (2010; 2011) reported that when compared to RC and A, they presented a similar clinical performance for all criteria, except for secondary carious lesions, in which GIC presented superior performance [30,33,35]. Although, there are two reviews reporting worse results for GICs performance. In the review by van’t Hof et al. (2006), it was reported that despite the high survival rates for single-surface ART restorations with HVGIC, survival rates for multi-surface ART restorations were low [28]. Similarly, Chisini et al. (2018) concluded that MRGIC exhibited the highest failure rate [8].

#### 3.5.4. Compomer (CO)

The efficacy of compomers as restorative materials was reported in six reviews [8,26,27,31,34,37], with calculated failure rates differing slightly for 24 and 36-month follow-ups (19% and 13%, respectively). Based on a moderate level of evidence, there was no statistically significant difference calculated between compomer and both conventional restorative materials and novel approaches, underlying that there is no advantage among these materials. In the study by Siokis et al. (2021), where meta-analysis was performed comparing compomer with RC and RMGIC, there were no statistically significant differences reported between the materials based on the “moderate” quality of evidence (RR 1.12 [0.41, 3.02]; *p* = 0.83; I^2^ = 57%, RR 1.04 [0.59, 1.84]; *p* = 0.88; I^2^ = 1%, respectively) [37]. In the review by Tedesco et al. (2018), rank probability, calculated through network meta-analysis, showed that the best results for treatment of occlusal caries are expected using compomer [34]. In the same study, regarding the treatment of multi-surface caries, compomers were ranked third after the Hall technique and non-restorative caries treatment.

#### 3.5.5. Stainless Steel Crowns (SSC)

Three reviews reported on the use performance of crowns for the restoration of carious primary teeth [8,29,34], with only one reporting calculable data. The failure rate was as low as 1% at 24 months and 4% at 36 months, underlying the excellent performance of the restorative material when compared to common conventional restorative materials. A meta-analysis reported that crowns have a reduced risk of a major failure at 24 months when compared to common filling materials (RR 0.18, 95% CI [0.06, 0.56], I^2^ = 0%) based on high-quality evidence [29].

#### 3.5.6. Secondary Outcomes

Data on secondary outcomes was non-existent in most cases, as quantitative synthesis in almost all the reviews reported effectiveness in favour of specific restorative materials. In the review by Innes et al. (2015) [29], it was found that crowns were less likely to cause pain than conventional restorations at 12–24 months (RR 0.15, 95% CI [0.04, 0.67]; I^2^ = 0%). Furthermore, participants reported more discomfort when a restoration was placed compared to crown placement (RR 0.56, 95% CI [0.36, 0.87]; I^2^ = 0%). The discomfort was defined as ‘moderate’, ‘intense’, or ‘very intense’ pain reported by children and ‘moderate’ or ‘significant’ patient discomfort rated by the dentist during treatment [29].

#### 3.5.7. Factors Affecting the Outcome

Different study designs and different handling of various materials regarding the necessity for application of local anaesthesia and/or isolation restricted the ability to assess the effect of the tooth and procedure-related factors on the outcome. The effect of single- or multi-surface restorations was demonstrated in four reviews [8,27,28,36], with most indicating that single-surface restorations exhibited lower failure rates as compared to multi-surface restorations (4.78%–7.6% vs. 9.46%–14.7%). Although, one review reported no difference in the weighted mean survival percentage of single- and multi-surface restorations [36]. Main reasons for the failure of both Class–I and Class–II restorations reported were secondary caries, restoration loss, and chipping of the marginal ridge with approximal contact loss. 

Regarding the effect of tooth isolation on the outcome, this was only reported in two studies [8,30], indicating higher success rates in restorations placed with rubber dam isolation (93.6% vs. 77.5%). In the review by Dias et al. (2018), it was shown that in procedures performed using cotton roll isolation, there was no significant difference between materials for all parameters analysed [30].

## 4. Discussion

This umbrella review reported on the clinical effectiveness of different restorative materials, including new biomaterials, for the restoration of carious lesions in primary teeth. Fourteen systematic reviews were included, revealing a total of 101 initial studies, the majority of which were RCTs. A variety of restorative materials (i.e., A, different GICs, CO, RC, and SSC) was used to restore single- or multi-surface carious lesions of affected primary teeth among the included primary studies. Moreover, two different restorative techniques were investigated, namely the conventional restorative treatment [8,20,26,27,30,31,32,33,34,35,36,37] and the atraumatic restorative treatment (ART) [20,27,28,33,34,35,36]. 

The overlap of primary studies calculated by the CCA was 0.06 (6%) revealing a moderate risk of bias due to inadvertently including the results of primary studies more than once [23]. Results showed a wide range of failure rates with all materials performing similarly and being equally efficient for restoration of primary teeth with deep caries. Amalgam and resin composite had the lowest mean failure rate at 24 months, while RMGIC and MRGIC had the highest, respectively, at 36 months. HVGIC showed the highest failure with compomer showing the lowest. Main reasons for the failure of both Class–I and Class–II restorations reported were secondary caries, restoration loss, and chipping of the marginal ridge with approximal contact loss. Evidence from reviews reporting on factors affecting the outcome indicated that single-surface restorations and restorations placed using rubber dam isolation exhibited lower failure rates. 

Conventional restorative materials, such as amalgam and resin composites, have a limited application in primary teeth. Despite their acceptable annual failure rate, their use in everyday practice is reduced. This can be attributed to patient-related factors that directly affect the execution of the restoration and therefore its longevity. Composite restorations are highly sensitive to the lack of complete moisture control, especially due to difficulties in cooperation, which can jeopardise the good performance of the material [8]. This underlines the increased prevalence of lost restorations reported in many studies. Similarly, amalgam, although having high durability with a survival range of 3.5 to 7 years, raises concern for its future use as a restorative material for Class–II cavities in primary teeth due to its toxicity and aesthetics [26]. Despite being considered an appropriate treatment option with high effectiveness and durability, its consideration as an option in future RCTs is questionable.

Less technique-sensitive materials are gaining interest and indicate high success rates in recent clinical studies. Their increased success rates are attributed to their biocompatibility and their easier and faster application, as compared to resin composites. It has been reported that the adhesion to the tooth structure is comparable to the micromechanical retention achieved by resin composite, giving the materials similar longevity rates [30]. Although, adhesion should not be the only criterion for retention of the restorative material as other factors directly related to the cavity (e.g., size, surfaces involved) may affect failure rates. At the same time, the less time-consuming application improves the procedure’s acceptability by the patients and has a positive effect on behavioural shaping and overall management of even uncooperative patients [39]. Results from previous studies reported similar results regarding the annual failure rate of GIC combined with ART and conventional restoration with composite or amalgam [27]. 

Our study indicated an equally excellent performance of GIC with improved materials, such as RMGIC, with better physical properties, presenting better fracture and wear resistance [17]. Results from most studies reported better performance of the latter for the restoration of small to moderate-sized proximal cavities. It is notable that in a recent study, a more than 5-times higher risk of failure was found when GIC was used for the restoration of Class–II cavities as compared to RMGIC restorations [40]. There are also studies reporting that the RMGIC survival rate is better than RC but worse than compomer [27]. This was mainly attributed to the worse surface roughness, anatomic form, and marginal adaptation of RMGICs. These shortcomings, in combination with the deficiency of a good colour match, limit the overall use of the material. 

GICs tend to form a stronger chemical bond, have a thermal expansion coefficient comparable to dentine, and release fluoride [17]. It has been suggested that fluoride released by these materials reduces tissue demineralisation and prevents caries recurrence, although the evidence is not strong as studies are not of high quality [41]. This is not in accordance with the results of other reviews that reported secondary caries as the main reason for the failure of GIC restorations, underlining that fluoride release does not affect the longevity of the material [8]. 

Evidence supports the use of simpler, less time-consuming techniques in a controlled environment for the treatment of deep caries in primary teeth. Simplification of adhesive systems and the use of flowable materials that are easily handled to achieve adaptation have been featured in RCTs with promising results. The limited lifespan of primary teeth favours their application, despite their questionable mechanical properties and shrinkage. However, the evidence to date is too limited to draw specific conclusions and further studies are required to investigate their efficacy.

In a similar way, bulk-fill composites are reliable materials for restoring primary teeth, which can decrease working time as they can fill the cavity in one step without layering. Studies comparing the material with other innovative techniques, such as reinforced GICs, confirmed their higher clinical success rate [32]. Compared to conventional restorative materials, they seem to have similar clinical outcomes, especially when referring to longevity. 

Excellent performance with very low failure rates was also observed for SSC based on high-quality evidence. Despite their reliability and longevity, they present a much more invasive treatment option when compared to other restorative materials commonly used. Additionally, restorative options after failure are limited due to the increased hard tissue removal, while common restorative materials could be replaced even after further tissue loss due to secondary caries. This highlights that the success rate should not be the only factor to be considered for the ideal choice of material in each case.

Our review supports that the failure rate of restorations is affected by tooth and technique-related factors. It has been demonstrated in almost all the included reviews that failure rates drop when materials are used in single-surface restorations [8,27,28]. Similarly, the use of rubber dam isolation increases the longevity of the restorations as compared to cotton roll isolation [8,30]. Despite the fact that it restricts saliva contamination and improves moisture control, therefore improving adhesive properties, it is not always easily applicable. This underlines the fact that having ideal application conditions in children is uncertain and therefore the choice of restorative material should be made carefully.

### 4.1. Strengths and Limitations

This umbrella review critically appraises the accessible evidence and presents a comprehensive overview of current restorative treatment approaches for carious lesions in primary teeth, which is its principal strength. The 14 included systematic reviews presented a broad spectrum of restorative treatment approaches for carious primary teeth, which have not been previously covered by one systematic review exclusively.

However, there are limitations of this umbrella review that need to be addressed. First and foremost, several review authors mentioned the heterogeneity among the included primary studies [20,29,31,33,36]. As far as the study design is concerned, five systematic reviews included only RCTs [8,27,28,32,34] and the included RCTs were further subdivided into split-mouth and parallel-arm studies. Pires et al. (2018) reported on a heterogeneity with regard to the sample size, the type of cavity and isolation, the restorative materials and techniques under investigation, as well as the length of the follow-up [31]. The comparison of studies and the evaluation of outcomes were complicated by the fact that a broad spectrum of restorative materials was included, with a few being used in a limited number of initial studies. It was further hampered since, within the same restorative material class, material compositions or adhesive concepts may have been modified by manufacturers over time to improve the material properties. The influence of the operator on the results of the studies also did not play a role in the included reviews [7]. By using different evaluation criteria with various cut-off points to assess the longevity of restorations, the data become even more heterogeneous [8,32]. The heterogeneity observed among the different primary studies often impedes the direct comparison of results. The quality of evidence is further restricted given the finding that several systematic reviews performed meta-analyses based on primary studies with an overall unclear or high risk of bias [20,30,31,34,35,37].

Secondly, a limit in the dropout rate for primary studies included in the systematic reviews was not chosen as an eligibility criterion for this umbrella review because of the wide variation of numbers presented within the included systematic reviews. Kilpatrick and Neumann (2007) observed a high dropout rate among primary studies on the use of amalgam in Class–II cavities in primary molars. In addition to this, inconclusive reporting of the results and different handlings when dealing with censored data, in case of restorations lost to follow-up or exfoliated teeth, hampered the calculation of failure rates [26].

Thirdly, the follow-up time among included primary studies was variable, ranging from 6 to 60 months. On the one hand, the lifespan of primary teeth in the oral cavity is limited because of the physiological exfoliation [8,26,32,34] which limits the maximum follow-up period, especially when older children are included. On the other hand, the occurrence of secondary caries at the restoration margins takes time, which is why its observation requires studies with a long-term follow-up [42,43].

Finally, the blinding of outcome assessors is often problematic, if not impossible, when restorative materials of different clinical appearances are compared with each other or with preformed crowns [20,29,31,36]. Therefore, the potential risk of bias in the measurement of the outcome has to be taken into consideration when interpreting the outcome of these studies.

The heterogeneity among included systematic reviews presented different study designs, several comparisons, and various outcome measures that precluded the quantitative synthesis of results. In five systematic reviews, meta-analyses were not performed and another two of the systematic reviews conducted network meta-analyses which did not allow for a comparison with conventional meta-analyses. Non-independence of samples was rarely assessed and causes of variation among included studies were not always discussed. For included studies with split-mouth design, cluster-level analysis and analysis of clustering effects were not reported. Ultimately, the unclear or high risk of bias among the majority of included systematic reviews ruled out meta-analyses.

### 4.2. Recommendations for Future Research

There is a need for further well-designed RCTs to overcome the limitations of studies on the restorative treatment of carious primary teeth and to increase the internal validity of future ones as well as the systematic and umbrella reviews [8,30,31,32,33,35,37]. This requirement was also observed in studies on (non-)vital pulp therapy in primary teeth [44]. In general, it is recommended to register the trial protocol from the start [45] and to follow the Consolidated Standards of Reporting Trials (CONSORT) to increase the transparency and completeness of reporting [33,35,45,46]. The trials should be conducted in parallel-arm design to avoid the shortcomings of split-mouth studies, with adequate random sequence generation and allocation sequence concealment to avoid bias arising from the randomisation process [35].

Future sample size calculations based on power analyses should take the high dropout rate observed among primary studies into account to obtain meaningful results after longer follow-up periods [47]. The inclusion of younger participants with a narrower age range would support the longer follow-up periods that are needed, given the shorter lifespan of primary teeth in the oral cavity leading to their physiological exfoliation [29,47]. In this respect, the handling of exfoliated teeth, censored data, and dropouts needs to be clarified, as the number of restorations at the different follow-ups affects the calculation of failure rates [47].

The caries risk of participants should be reported, as it was shown that a higher caries risk is associated with increased susceptibility to restoration failure [43,48]. The fact that a secondary caries was found to be the main reason for failure in primary [8] and permanent teeth [43,48] shows that the participants are at risk of caries even after the restorative treatment and illustrates the need for behavioural changes by employing additional caries-preventive strategies to achieve a long-term success [8]. Innes et al. (2015) recommended extending the inclusion criteria of future studies to children with special needs or developmental defects of teeth and to include general dental practitioners without specialisation in an attempt to increase the generalisability of the outcome [29]. 

As far as the teeth are concerned, the type of included teeth and the cavity class should be mentioned. This is connected to the extension of the carious lesions, which were frequently found to be stated insufficiently [29,36]. Therefore, the carious lesion depth and its impact on the outcome should be reported [29].

The experience of the operators performing the restorative treatment should be mentioned because it may influence the clinical performance of restorations [27,48]. The reporting of treatment-related factors, such as the choice of anaesthesia, the isolation technique, and detailed descriptions of restorative materials and techniques facilitates the interpretation of results, which is desirable for future studies.

For the outcome assessment, validated and internationally accepted criteria should be used to determine the restorative treatment success. The clinical criteria introduced by the FDI World Dental Federation in 2007 have been used in an increasing number of clinical trials and may be a viable option for further studies [49,50]. In addition to the classical outcome measures, patient-related and -reported factors should be further documented, such as discomfort, pain, and the impact on the oral health-related quality of life [29,47].

## 5. Conclusions

Within the limitations of the current review, the conclusions drawn were:All restorative materials have acceptable mean failure rates and their use for the treatment of carious primary teeth is supported.Among common conventional restorative materials, amalgam and resin composite had the lowest mean failure rates at 24 months and compomer at 36 months.Stainless steel crowns had the lowest failure rate at 24 months and 36 months compared to all other restorative materials for primary teeth.Limited reviews indicated that single-surface restorations and restorations placed using rubber dam isolation exhibited lower failure rates.The main reasons reported for the failure of both Class–I and Class–II restorations were secondary caries, restoration loss, and chipping of the marginal ridge with approximal contact loss.

## Figures and Tables

**Figure 1 jcm-11-03490-f001:**
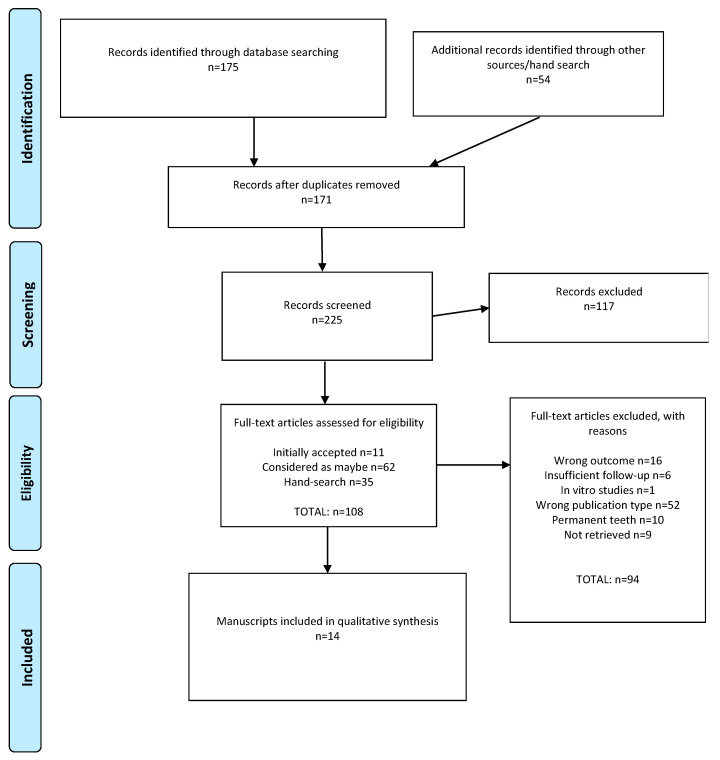
PRISMA flow diagram.

**Figure 2 jcm-11-03490-f002:**
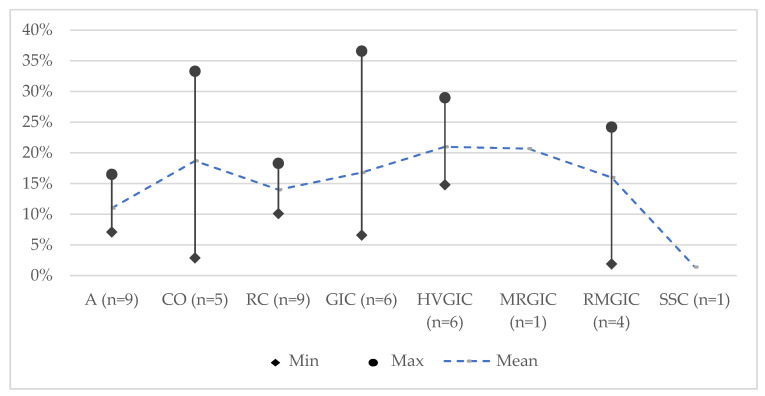
The failure rate of restorative materials at 24 months. Abbreviations: A: amalgam; CO: compomer; GIC: glass-ionomer cement; HVGIC: high-viscosity glass-ionomer cement; MRGIC: metal-reinforced glass-ionomer cement; RC: resin composite; RMGIC: resin-modified glass-ionomer cement; SSC: stainless steel crown; n: represents the number of reviews in which the efficacy of the specific material was assessed; Min: minimum value for failure rate; Max: maximum value for failure rate.

**Figure 3 jcm-11-03490-f003:**
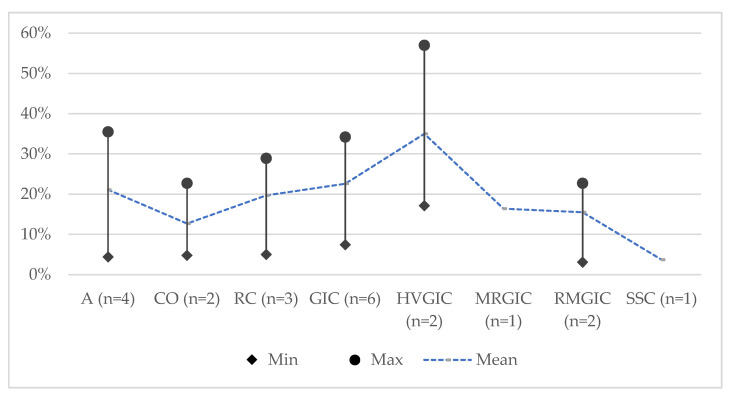
The failure rate of restorative materials at 36 months. Abbreviations: A: amalgam; CO: compomer; GIC: glass-ionomer cement; HVGIC: high-viscosity glass-ionomer cement; MRGIC: metal-reinforced glass-ionomer cement; RC: resin composite; RMGIC: resin-modified glass-ionomer cement; SSC: stainless steel crown; *n*: represents the number of reviews in which the efficacy of the specific material was assessed; Min: minimum value for failure rate; Max: maximum value for failure rate.

**Table 1 jcm-11-03490-t001:** Main characteristics of the included studies.

Author, Year	Total Number and Study Designs Included	Intervention	Comparator Group	Outcomes	Criteria	Follow-Up (Months)	Quantitative Synthesis Performed
Siokis et al., 2021 [37]	10 RCTs (5 split-mouth and 4 parallel-arm)	Tooth-coloured materials	Between each other	Failure rate	NR	18–48	Six studies included
Chisini et al., 2018 [8]	17 RCTs (8 split-mouth and 5 parallel-arm); 14 non-RCTs (6 parallel-arm and 1 split-mouth)	Conventional restorative materials (A, GIC, CO, RC, MRGIC, RMGIC,) and SSC	Materials, techniques, and related factors associated with restoration failure	Annual failure rate, survival rate, and success rate	Modified USPHS (*n* = 21); FDI (*n* = 2); own criteria (*n* = 8)	12–48	Not performed
Delgado et al., 2021 [32]	7 RCTs (6 split-mouth); 1 non-RCT	RC which varied the resin-based composite, underlying adhesive strategy, or the application strategy/mode	Between each other	Survival rate and retention	Modified USPHS (*n* = 6); FDI (*n* = 2)	12–36	Not performed
Dias et al., 2018 [30]	10 RCTs (6 split-mouth and 4 parallel-arm)	GIC and RMGIC	RC	Clinical performance (secondary caries, marginal discolouration/adaptation, longevity, retention, wear, and anatomical form)	Modified USPHS (*n* = 7); FDI (*n* = 2); own criteria (*n* = 1)	6–48	Nine studies included
Frencken et al., 2021 [36]	6 RCTs (3 split-mouth and 3 parallel-arm)	Combination of ART and HVGIC	A and RC	Survival rate	ART (*n* = 4); USPHS (*n* = 1); ART/USPHS (*n* = 1)	24–36	Not performed
Innes et al., 2015 [29]	5 RCTs (3 split-mouth and 1 parallel-arm)	Preformed crowns	Conventional restorative materials	Major failure	NR	12–60	Three studies included
Kilpatrick et al., 2007 [26]	17 RCTs (11 split-mouth and 5 parallel-arm)	A	CO, RC, and GIC	Failure rate	ART (*n* = 1); USPHS (*n* = 10); ART/USPHS (*n* = 1); DPDHS (*n* = 1)	24–96	Not performed
Mickenautsch et al., 2009 [38]	3 RCTs (2 split-mouth and 1 parallel-arm)	ART using GIC	A	Longevity (dichotomous success/failure rates)	ART (*n* = 3)	12–36	Two studies included
Mickenautsch et al., 2011 [35]	6 RCTs (2 split-mouth, 1 parallel-arm, and 3 partial split-mouth)	GIC	A	Recurrent caries, caries on margins, and caries progression	ART (*n* = 2); USPHS (*n* = 3); DPDHS (*n* = 1)	12–60	Two studies included
Pires et al., 2018 [31]	17 RCTs (10 split-mouth, 1 split-mouth in most samples, and 6 parallel-arm)	Conventional restorative materials (A, CO, RC, GIC, RMGIC, HVGIC, and MRGIC)	Between each other	Survival rate	USPHS (*n* = 15); FDI (*n* = 2)	12–60	Seventeen studies included
Ruengrungsom et al., 2018 [27]	32 RCTs (13 split-mouth and 19 parallel-arm); 3 retrospective studies	GIC (ART and conventional) restorations	Other tested materials	AFR and qualitative description (five studies)	(Modified) USPHS (*n* = 15); ART (*n* = 10); ART/USPHS (*n* = 2); FDI (*n* = 2); Roeleveld (*n* = 2); Gemert–Schrik’s criteria (*n* = 1); own criteria (*n* = 8)	18–84	Not performed
Tedesco et al., 2017 [20]	4 RCTs (2 split-mouth and 2 parallel-arm)	ART restorations with HVGIC	Conventional Class–II restorations with A and RC	Longevity, pulp damage, and caries lesion progression	Modified USPHS (*n* = 1); ART (*n* = 3)	24–36	Four studies included
Tedesco et al., 2018 [34]	14 RCTs (5 split-mouth and 9 parallel-arm); 1 observational study	CRT, ART, and HVGIC	Between each other	Success rate and caries lesion arrestment	ART (*n* = 4); ART and USPHS (*n* = 1); USPHS (*n* = 1); criteria by Innes et al., 2007 (*n* = 2); criteria by Aguilar et al., 2007 (*n* = 1); criteria by Houpt et al., 1983 (*n* = 1); based on Miller, 1959 and Kidd, 2010 (*n* = 1); PUFA-Index (*n* = 1); visual and tactile characteristics of caries lesion arrestment (*n* = 2); according to the dentist’s assessment (*n* = 1)	6–84 s	Thirteen studies included
van’t Hof et al., 2006 [28]	7 RCTs (3 split-mouth and 4 parallel-arm); 1 longitudinal; and 1 NR	ART restorations using medium and high-viscosity GIC	Between each other	Success rate and mean AFR	Most used ART criteria	12–36	Ten studies included

Abbreviations: A: Amalgam; AFR: annual failure rate; ART: atraumatic restorative treatment; CO: compomer; CRT: conventional restorative treatment; DPDHS: Danish Public Dental Health Service criteria; GIC: glass-ionomer cement; FDI: World Dental Federation; HVGIC: high-viscosity glass-ionomer cement; MRGIC: metal-reinforced glass-ionomer cement; NR: not reported; PUFA: index of clinical consequences of untreated dental caries (pulpal involvement/ulceration/fistula/abscess); RC: resin composite; RCT: randomized controlled trial; RMGIC: resin-modified glass-ionomer cement; SSC: stainless steel crown; USPHS: United States Public Health Service criteria.

**Table 2 jcm-11-03490-t002:** Quality assessment of included reviews using the ROBIS tool.

Author, Year	Quality Assessment Tool Used	Review Process	Risk of Bias	Concerns
SEC	ISS	DCSA	SF
Siokis et al., 2021[37]	Cochrane Collaboration tool						Meta-analysis performed using studies of unclear or high risk of bias
Chisini et al., 2018[8]	Cochrane Collaboration tool						Language restrictions
Delgado et al., 2021[32]	Cochrane Collaboration tool						Publication date restrictions
Dias et al., 2018[30]	Cochrane Collaboration tool						Inclusion criteria and outcomes not clearly mentioned, a meta-analysis performed using studies of high risk of bias
Frencken et al., 2021[36]	Criteria by De Amorim et al., 2017			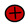		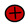	Search strategy not reproducible; inclusion criteria not clearly mentioned; outcomes not pre-defined; no details given on data collection and appraisal; no clear data synthesis; and risk of bias tool not validated
Innes et al., 2015[29]	Cochrane Collaboration Tool						—
Kilpatrick et al., 2007[26]	Criteria by Curzon and Toumba, 2006			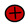		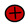	Restricted data search; language restrictions; risk of bias tool not validated; no evaluation of clinical and methodological heterogeneity; and limited data on data collection and synthesis
Mickenautsch et al., 2010[33]	Cochrane Collaboration tool						Language restrictions
Mickenautsch et al., 2011[35]	Based on the availability of evidence indicating successful prevention of selection and detection/performance bias from start to the end of each trial						Risk of bias tool not validated; meta-analysis performed using studies of unclear or high risk of bias
Pires et al., 2018[31]	Cochrane Collaboration Tool						Meta-analysis performed using studies of unclear or high risk of bias
Ruengrungsom et al., 2018[27]	Cochrane Collaboration Tool						Restricted search strategy and data search; language restrictions; no details given on data collection and appraisal; primary studies with clinical and methodological heterogeneity; a majority of studies with an unclear risk of bias; and no meta-analysis performed
Tedesco et al., 2017[20]	Cochrane Collaboration Tool						Meta-analysis performed using studies of unclear risk of bias; heterogeneity and publication bias of primary studies
Tedesco et al., 2018[34]	Cochrane Collaboration Tool						Meta-analysis performed using studies of unclear or high risk of bias
van’t Hof et al., 2006[28]	NR		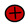	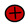		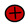	Language restrictions; search strategy not reproducible; restricted data search; inclusion criteria not clearly mentioned; and risk of bias tool not reported

Abbreviations: SEC: study eligibility criteria; ISS: identification and selection of studies; DCSA: data collection and study appraisal; SF: synthesis and findings. 

 = Low risk of bias. 
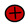
 = High risk of bias. 

 = Unclear risk of bias.

**Table 3 jcm-11-03490-t003:** Main conclusions drawn from the included reviews.

Author, Year	Restorative Material Assessed (Number of Studies)	Type of Teeth	Number of Teeth Assessed (Initially/Latest Follow-Up)	Type of Cavity	Isolation	Caries Removal Technique	Conclusions
Siokis et al., 2021 [37]	CO (*n* = 4); GIC (*n* = 3); HVGIC (*n* = 1); RC (*n* = 8); RMGIC (*n* = 7)	Posterior	1023 restorations	Class–II	NR	NR	Resin-based restorative materials (RC and CO) and RMGIC appeared to have no statistically significant differences based on a “moderate” level of evidence.
Chisini et al., 2018 [8]	A (*n* = 6); CO (*n* = 9); GIC (*n* = 5); MRGIC (*n* = 4); RC (*n* = 6); RMGIC (*n* = 10); SSC (*n* = 3)	Posterior	8679/7392, range: 40–1834	Class–I (*n* = 1); Class–II (*n* = 15); Class–I/II (*n* = 12); crown restorations (*n* = 3)	RD (*n* = 11); no isolation (*n* = 10); yes/no (*n* = 3)	NR	Resin composite exhibited the lowest failure rates, whereas MRGIC exhibited the highest. SSC had the highest success rate. Higher success rates were observed in restorations of a single tooth surface and those performed with rubber dam isolation. Secondary caries was the main reason for failure.
Delgado et al., 2021 [32]	RC (*n* = 5); adhesive systems (*n* = 2); surface pre-treatments (*n* = 1); reducing etching time of an etch-and-rinse contemporary adhesive (*n* = 1); novel self-adhesive composites (*n* = 2); application modes of a universal adhesive (*n* = 1); bulk-fill RC (*n* = 1); sonic-resin placement system in bulk (*n* = 1); novel base RC (*n* = 1); contemporary adhesives (*n* = 1)	NR	723 restorations	Class–I (*n* = 494); Class–II (*n* = 229)	RD (*n* = 6)	CCR (*n* = 7); SCR (*n* = 1)	Novel approaches such as bulk-fill resin composites, self-adhesive restoratives, and adhesives have comparable performance to traditional materials. All materials were deemed clinically acceptable in children.
Dias et al., 2018 [30]	RMGIC (*n* = 6); GIC (*n* = 4); RC (*n* = 10)	Posterior	1425/932, range:75–344/8–207	Class–II	RD (*n* = 7); CR (*n* = 3)	NR	GIC and RC presented a similar clinical performance for all criteria analysed, except for secondary carious lesions in which GIC presented superior performance, especially for the RMGIC and with rubber dam isolation.
Frencken et al., 2021 [36]	A (*n* = 4); GIC (*n* = 6); RC (*n* = 2)	Posterior	2067 restorations	Class–I/II (*n* = 6)	NR	NR	No statistically significant difference was found between the weighted mean survival percentages of ART/HVGIC and traditional treatments in both single- and multiple-surface restorations in the primary molars.
Innes et al., 2015 [29]	PMC (*n* = 5); open sandwich using RMGIC or RC (*n* = 2); restorative materials (*n* = 2); aesthetic crown (*n* = 1); non-restorative treatment (*n* = 1)	Posterior	80–264 teeth	Multiple-surface	NR	CCR +/− PCR (*n* = 1); pulpotomy (*n* = 2)	Crowns placed on primary teeth with carious lesions reduce the risk of major failure or pain in the long term compared to fillings.
Kilpatrick et al., 2007 [26]	A (*n* = 17); CO (*n* = 8); GIC (*n* = 8); RC (*n* = 3)	Posterior	Range: 40–1035	Class–II	RD (*n* = 8); no RD (*n* = 5)	NR	Amalgam used to restore interproximal (Class–II) cavities in primary molars can be expected to survive a minimum of 3.5 years, but potentially in excess of 7 years, remains an appropriate treatment option for the management of caries in children.
Mickenautsch et al., 2010 [33]	A (*n* = 3); HVGIC (*n* = 3)	Posterior	1951 restorations at latest follow-up, range: 5–610	Class–I (*n* = 1); Class–I/II (*n* = 2)	NR	NR	ART restorations with HVGIC appear to be equally successful, and their survival rate may even exceed that of amalgam fillings.
Mickenautsch et al., 2011 [35]	GIC (*n* = 9); A (*n* = 9)	Posterior	Range: 32–1035 teeth	Single-surface (*n* = 3); multiple-surface (*n* = 2); combination (*n* = 4)	NR	NR	GIC-restored cavities show less recurrent carious lesions than cavities restored with amalgam.
Pires et al., 2018 [31]	A (*n* = 8); CO (*n* = 9); GIC (*n* = 3); RC (*n* = 10); RMGIC (*n* = 7)	Posterior	2687 teeth	Class–I/II (*n* = 7); Class–II (*n* = 10)	RD (*n* = 9); no RD (*n* = 6); NR (*n* = 2)	NR	There is no advantage among restorative treatments using CO, RMGIC, A, and RC, although GIC conventional restorations have a higher risk of failure.
Ruengrungsom et al., 2018 [27]	A (*n* = 9); CO (*n* = 4); GIC (*n* = 7); Giomer (*n* = 1); HVGIC (*n* = 23); RC (*n* = 8); RMGIC (*n* = 13); MRGIC (*n* = 4)	Posterior	Conventional restorations: 3976/3381, range: 13–456; ART restorations: 6959/4588, range: 13–425	Class–I; Class–II; multiple-surface	RD (*n* = 10); partially RD (*n* = 1); no RD (*n* = 2); NR (*n* = 27)	NR	The conventional technique showed a higher survival rate than ART for Class–I and multi-surface restorations with GIC. For both restorative approaches, the AFRs of Class–II and multi-surface GIC restorations were increased compared to those of Class–I restorations. The main reasons for the failure of Class–I and Class–II restorations were restoration loss and chipping of the marginal ridge with approximal contact loss.
Tedesco et al., 2017 [20]	A (*n* = 3); HVGIC (*n* = 4); RC (*n* = 1)	Posterior	ART restorations: 985/NR, range: 9–610; conventional restorations: 786/NR, range: 9–425	Class–II	NR	Spoon excavator (ART); NR (conventional)	ART Class–II restorations with HVGIC presented a similar survival rate compared to conventional Class–II restorations with RC/A.
Tedesco et al., 2018 [34]	A (*n* = 4); CO (*n* = 2); HT (*n* = 2) HVGIC (*n* = 8); NR (*n* = 1); NRCT (*n* = 1); RC (*n* = 3); RMGIC (*n* = 1); RS (*n* = 2); SSC (*n* = 1); SDF (*n* = 3); UCT (*n* = 1)	Posterior (probably)	8064 teeth, range: 9–1107	Class–I (*n* = 11); Class–II (*n* = 10); smooth surface (*n* = 3)	NR	Hand instrument (ART); rotary (conventional)	CRT with resin composite demonstrated better performance compared to resin sealant.
van’t Hof et al., 2006 [28]	GIC (*n* = 1); HVGIC (*n* = 8)	NR (probably posterior)	NR	Single-surface; multiple-surface	NR	NR	While single-surface ART restorations with HVGIC exhibited high survival rates, those of multi-surface ART restorations were low.

Abbreviations: A: Amalgam; AFR: annual failure rate; ART: atraumatic restorative treatment; CCR: complete caries removal; CO: compomer; CR: cotton rolls; CRT: conventional restorative treatment; GIC: glass-ionomer cement; HT: Hall technique; HVGIC: high-viscosity glass-ionomer cement; MRGIC: metal-reinforced glass-ionomer cement; NR: not reported; NRCT: non-restorative caries treatment; PCR: partial caries removal; PMC: preformed metal crown; RC: resin composite; RMGIC: resin-modified glass-ionomer cement; RD: rubber dam; RS: resin sealing; SCR: selective caries removal; SDF: silver diamine fluoride; SSC: stainless steel crown; UCT: ultraconservative treatment.

**Table 4 jcm-11-03490-t004:** Main results from the meta-analysis of included reviews with a quantitative synthesis of the results.

Review	Restorative Material				
	RC	GIC	HVGIC	RMGIC	Crowns
Siokis et al., 2021 [37]	RC vs. CO: RR 1.12 (0.41, 3.02); *p* = 0.83; I^2^ = 57% RC vs. RMGIC: RR 1.10 (0.74, 1.63); *p* = 0.65; I^2^ = 0%			RMGIC vs. CO: RR 1.04 (0.59, 1.84); *p* = 0.88; I^2^ = 1%	
Dias et al., 2018 [30]	CO vs. GIC: Overall: RR 0.03 (–0.00, 0.06); *p* = 0.06; I^2^ = 27% Marginal adaptation: RR 0.00 (–0.05, 0.05); *p* = 1.00; I^2^ = 0% Marginal discoloration: RR 0.07 (–0.08, 0.21); *p* = 0.38; I^2^ = 77% Anatomic form: RR 0.01 (–0.03, 0.06); *p* = 0.58; I^2^ = 0% Secondary caries: RR 0.06 (0.02, 0.10); *p* = 0.008; I^2^ = 0%				
Innes et al., 2015 [29]					Crowns vs. fillings: Major failure: RR 0.18, 95% CI (0.06, 0.56), I^2^ = 0% at 24 months Crowns vs. NRCT: Major failure: RR 0.12; 95% CI (0.01, 2.18) at 12 months
Mickenautsch et al., 2009 [38]		GIC vs. A: OR 2.00; CI (0.06–5.05); *p* = 0.10 at 36 months			
Mickenautsch et al., 2010 [33]			GIC vs. A: RR 0.93; 95% CI (0.83, 1.06); *p* = 0.26 at 12 months RR 1.07; 95% CI (0.91, 1.27); *p* = 0.39 at 24 months		
Pires et al., 2018 [31]		GIC vs. RC: RR 4.00; 95% CI (1.19, 13.41) GIC vs. RMGIC: RR 4.70; 95% CI (1.09, 20.27) GIC vs. A: RR 1.62; 95% CI (1.05, 2.52)		RMGIC vs. A: RR 0.6; 95% CI (0.42, 0.86)	
Tedesco et al., 2017 [20]			Pooled estimate for ART success: OR 0.887, 95% CI (0.574, 1.371)		
Tedesco et al., 2018 [34]	RC vs. RS: Overall: RR 11.16, 95% CI (2.46, 50.62) Caries arrest: RR 7.89, 95% CI (0.39, 160.91)				
van’t Hof et al., 2006 [28]			Weighted mean % for survival (95% CI) for single-surface ART: 12 months 95 (94, 97) 24 months 91 (88, 93) 36 months 86 (83, 90). Weighted mean % for survival (95% CI) for multiple-surface ART: 12 months 73 (70, 77) 24 months 59 (55, 64) 36 months 49 (44, 54)		

Abbreviations: A: Amalgam; ART: atraumatic restorative treatment; CI: confidence interval; CO: compomer; GIC: glass-ionomer cement; HVGIC: high-viscosity glass-ionomer cement; NRCT: non-restorative caries treatment; OR: odds ratio; RC: resin composite; RMGIC: resin-modified glass-ionomer cement; RR: risk ratio; RS: resin sealing.

## Data Availability

The data presented in this umbrella review are available in the manuscript and the Appendix A. Further information is available on request from the corresponding author.

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
