# Peer review of "Clinical Effectiveness of Restorative Materials for the Restoration of Carious Primary Teeth: An Umbrella Review"

_jcm, 2022, doi:10.3390/jcm11123490_

Round 1
Reviewer 1 Report
Dear Authors,
I would like to congratulate for the detailed and excellent work. However, please let me raise a few questions and issues:
-I do find the introduction a bit too long, as the article has already an enormous length, I would suggest reducing the length of the introduction a bit, if possible.
-Tables in their current format make the article very difficult to read. Please make sure, after acceptance, that in the final format a more suitable layout can be presented (shorter or linked tables)
-I can totally agree with the findings of the stainless steel crowns and their reliability, however, I miss a bit from the discussion the basic fact, that a crown is much more invasive, so however its failure rate is lower, the restorative options are much less after a crown failure, than after failure of filings. This can be either added to discussion or even in the limitations part. I just do not want the readers to conclude, that they shall prefer crowns instead of fillings…
Author Response
Dear Reviewer 1,
We would like to thank you for the valuable and constructive comments. We have revised the manuscript based on the reviewers’ suggestions. Please find enclosed the revised manuscript with track changes turned on to spot the changes we have made in the manuscript. In addition to that we have revised the manuscript with regard to English language. Please also find our response to the reviewers addressing each comment in the following paragraphs. We think that the revisions made based on the reviewers’ comments have considerably improved the quality of the manuscript. We hope the revised manuscript is acceptable for publication in the Journal of Clinical Medicine.
Yours sincerely,
The review team
Response to Reviewer 1
- Comment Reviewer 1:
Dear Authors,
I would like to congratulate for the detailed and excellent work. However, please let me raise a few questions and issues:
-I do find the introduction a bit too long, as the article has already an enormous length, I would suggest reducing the length of the introduction a bit, if possible.
Response to Reviewer 1:
Thank you very much for this comment. We agree that the introduction was of considerable length. Therefore, with have deleted some sentences to shorten the introduction.
Revised text:
The following lines of the introduction have been deleted: Pg. 2, lines 62-64, 70-71, 77-79, 82-85.
- Comment Reviewer 1:
Tables in their current format make the article very difficult to read. Please make sure, after acceptance, that in the final format a more suitable layout can be presented (shorter or linked tables)
Response to the Reviewer:
We thank the reviewer for this comment. We followed the submission guidelines of the Journal of Clinical Medicine regarding the layout of the tables. We decided not to shorten the tables, since relevant information may have been lost for the reader otherwise. However, we will make sure that the tables will be presented in a more suitable layout after acceptance and will ask the editorial office for the possibility of linked tables.
Revised text:
Not applicable at the moment.
- Comment Reviewer 1:
I can totally agree with the findings of the stainless steel crowns and their reliability, however, I miss a bit from the discussion the basic fact, that a crown is much more invasive, so however its failure rate is lower, the restorative options are much less after a crown failure, than after failure of filings. This can be either added to discussion or even in the limitations part. I just do not want the readers to conclude, that they shall prefer crowns instead of fillings…
Response to the Reviewer:
We agree with the reviewer’s comment – thank you! We have revised the discussion part based on the reviewer’s suggestion.
Revised text:
Pg. 18, lines 442-448: “(…) Excellent performance with very low failure rates was also observed for SSC based on high quality evidence. Despite their reliability and longevity, they present a much more invasive treatment option when compared to other restorative materials commonly used. Also, restorative options after failure are limited due to the increased hard tissue removal, while common restorative materials could be replaced even after further tissue loss due to secondary caries. This highlights that success rate should not be the only factor to be considered for the choice of ideal material in each case. (…)”
Reviewer 2 Report
- Move calibration of reviewers section from results (line 194) to methods (line 153)
- Line 226, why did the authors not exclude studies that had RCTs and an observational/longitudinal study? please provide explanation within the text.
- In figure 2 and 3, what is A? what represents min and max?
- Section 2.5, When was the search performed? What years did the search include?
- Section 4.2, suggest future similar studies for root canal filling materials and techniques. As an example, cite the following article and other similar reviews. "Primary tooth pulpectomy overfilling by different placement techniques: A systematic review and meta-analysis.Journal of Dental Research, Dental Clinics, Dental Prospects 14.4 (2020): 250.”
- In discussion, talk about the duration of each filling option. The below references mentions that duration of treatment affect children’s behavior, consequently, the shorter filling and restoration time, might be better for behavioral management. Cite the following :
"Does the length of dental procedure influence children’s behavior during and after treatment? A systematic review and critical appraisal. JODDD 12.1 (2018): 68. “
"Barriers and drawbacks of the assessment of dental fear, dental anxiety and dental phobia in children: a critical literature review. Journal of Clinical Pediatric Dentistry 41.6 (2017): 399-423.”
“Use of general anaesthesia in paediatric dentistry: barriers to discriminate between true and false cases. European Archives of Paediatric Dentistry 17 (2), 89-95” - In conclusion, say “within the limitations of the current study”
Author Response
Dear Reviewer,
We would like to thank you for the valuable and constructive comments. We have revised the manuscript based on the reviewers’ suggestions. Please find enclosed the revised manuscript with track changes turned on to spot the changes we have made in the manuscript. In addition to that we have revised the manuscript with regard to English language. Please also find our response to the reviewers addressing each comment in the following paragraphs. We think that the revisions made based on the reviewers’ comments have considerably improved the quality of the manuscript. We hope the revised manuscript is acceptable for publication in the Journal of Clinical Medicine.
Yours sincerely,
The review team
Response to Reviewer 2
- Comment Reviewer 2:
Move calibration of reviewers section from results (line 194) to methods (line 153)
Response to the Reviewer:
We thank the reviewer for this suggestion and have added the calibration to the methods.
Revised text:
Calibration has been deleted from the results and added to the methods (Pg. 3, lines 147-149).
- Comment Reviewer 2:
Line 226, why did the authors not exclude studies that had RCTs and an observational/longitudinal study? please provide explanation within the text.
Response to the Reviewer:
Thank you for this question! The inclusion criteria were kept broadly to obtain a maximum number of eligible studies. Therefore, the study design was not a limitation for this umbrella review. Nevertheless, we are aware of the fact that results from well-designed RCTs are of highest evidence.
Revised text:
Pg. 3, line 99: “(…) of any design (…)”
- Comment Reviewer 2:
In figure 2 and 3, what is A? what represents min and max?
Response to the Reviewer:
We thank the reviewer for this question; “A” means amalgam, “min” means minimal value for failure rate, and “max” maximum value for failure rate. We have added this information to Figure 2 and 3.
Revised text:
Pg. 10, Figure 2, pg. 15, Figure 3: Abbreviations of Figure 2 and 3.
- Comment Reviewer 2:
Section 2.5, When was the search performed? What years did the search include?
Response to the Reviewer:
Thank you for these questions. The search was conducted on 7th October 2021. Neither language nor publication year restrictions were applied. We have added this information to the manuscript.
Revised text:
Pg. 3, line 134 and 139-140: “(…) (07.10.2021) (…) No language or publication year restrictions were applied. (…)”
- Comment Reviewer 2:
Section 4.2, suggest future similar studies for root canal filling materials and techniques. As an example, cite the following article and other similar reviews. "Primary tooth pulpectomy overfilling by different placement techniques: A systematic review and meta-analysis.Journal of Dental Research, Dental Clinics, Dental Prospects 14.4 (2020): 250.”
Response to the reviewer:
We express our gratitude to the reviewer for this constructive comment. This topic was beyond the scope of this review, as we focused on restorative materials for primary teeth without vital pulp therapy and non-vital pulp therapy. We will consider in future projects.
Revised text:
Not applicable.
- Comment Reviewer 2:
In discussion, talk about the duration of each filling option. The below references mentions that duration of treatment affect children’s behavior, consequently, the shorter filling and restoration time, might be better for behavioral management. Cite the following :
- "Does the length of dental procedure influence children’s behavior during and after treatment? A systematic review and critical appraisal. JODDD1 (2018): 68.”
- "Barriers and drawbacks of the assessment of dental fear, dental anxiety and dental phobia in children: a critical literature review. Journal of Clinical Pediatric Dentistry 41.6 (2017): 399-423.2”
- “Use of general anaesthesia in paediatric dentistry: barriers to discriminate between true and false cases. European Archives of Paediatric Dentistry 17 (2), 89-95”
Response to the Reviewer:
Thank you for this constructive comment. We have added information about the duration of the restorative procedure to the discussion and cited the reference mentioned by the reviewer fitting best to our discussion.
Revised text:
Pg. 17, lines 407-409: “(…) At the same time, the less time-consuming application improves the procedure’s acceptability by the patients and has a positive effect on behavioural shaping and overall management of even uncooperative patients [39]. (…)”
- Comment Reviewer 2:
In conclusion, say “within the limitations of the current study”
Response to the Reviewer:
Thank you very much – have have added this to the conclusion.
Revised text:
Pg. 20, line 553: “(…) Within the limitations of the current review, the conclusions drawn were: (…)”
Reviewer 3 Report
The review is well done.
I suggest to always try to draw conclusions from the review and in general from a scientific article.
Author Response
Dear Reviewer,
We would like to thank you for the valuable and constructive comments. We have revised the manuscript based on the reviewers’ suggestions. Please find enclosed the revised manuscript with track changes turned on to spot the changes we have made in the manuscript. In addition to that we have revised the manuscript with regard to English language. Please also find our response to the reviewers addressing each comment in the following paragraphs. We think that the revisions made based on the reviewers’ comments have considerably improved the quality of the manuscript. We hope the revised manuscript is acceptable for publication in the Journal of Clinical Medicine.
Yours sincerely,
The review team
Response to Reviewer 3
Comment Reviewer 3:
The review is well done.
I suggest to always try to draw conclusions from the review and in general from a scientific article.
Response to the Reviewer:
Thank you very much for your kind comment. We agree with your comment and have revised the conclusion based on your suggestion.
Revised text:
Pg. 1, lines 32-34: “(…) Within the limitations of the review, all materials have acceptable mean failure rates and could be recommended for the restoration of carious primary teeth. (…)”
Pg. 20, lines 554-564: “(…)
- All restorative materials have acceptable mean failure rates and their use for the treatment of carious primary teeth is supported.
- Among common conventional restorative materials, amalgam and resin composite had the lowest mean failure rates at 24 months and compomer at 36 months.
- Stainless steel crowns had the lowest failure rate at 24 months and at 36 months compared to all other restorative materials for primary teeth.
- Limited reviews indicated that single–surface restorations and restorations placed using rubber dam isolation exhibited lower failure rates.
- Main reasons for failure of both Class–I and Class–II restorations were secondary caries, restoration loss and chipping of the marginal ridge with approximal contact loss. (…)”
Round 2
Reviewer 2 Report
all comments are answered